# Business Results and Well-Being: An Engaging Leadership Intervention Study

**DOI:** 10.3390/ijerph17124515

**Published:** 2020-06-23

**Authors:** Lars van Tuin, Wilmar B. Schaufeli, Willem van Rhenen, Rebecca M. Kuiper

**Affiliations:** 1Social, Health and Organizational Psychology, Utrecht University, 3584 CS Utrecht, The Netherlands; w.schaufeli@uu.nl; 2Research Unit Work Occupational & Organizational Psychology and Professional Learning, KU Leuven, B-3000 Leuven, Belgium; 3Engagement & Productivity, Nyenrode Business Universiteit, 3621 BG Breukelen, The Netherlands; w.vrhenen@nyenrode.nl; 4Department of Methodology and Statistics, Utrecht University, 3584 CH Utrecht, The Netherlands; R.M.Kuiper@uu.nl

**Keywords:** co-creation, leadership development, self-determination theory, engaging leadership, intrinsic motivation, absenteeism, well-being

## Abstract

The present quasi-experimental study tested the business impact of a leadership development program focusing on psychological well-being through the satisfaction of basic psychological needs. Based on the concept of engaging leadership and self-determination theory, the 8-month program targeted midlevel team leaders of the customer fulfilment center of a health systems multinational organization. The program was designed in co-creation between senior leadership and the team leaders that participated in the program. Outcomes showed positive business results through significant increases in a preselected key performance indicator and decreased employee absenteeism. Through changes in autonomy satisfaction and intrinsic motivation, the team leaders (*N* = 14) benefitted in a moderate to very large extent relative to a similar control group (*N* = 52). In contrast, team members (*N* = 148) displayed no such benefits. Specifically, higher levels of autonomy satisfaction are said to lead to higher levels of psychological well-being and motivation. Still, the link with business performance is absent in most organizational studies within self-determination theory, making the present study one of the first to fill this gap. The study discloses the program design, compares the effects to a relevant control group, evaluates the lessons learned, and provides practical suggestions.

## 1. Introduction

Leadership effectiveness is of primary concern to organizations—hence the popularity of leadership development programs and the considerable investment organizations make in such programs [1]. Organizations are confronted with rapid change, growing interdependencies, and complexity in what is generally characterized as a world of volatility, uncertainty, complexity, and ambiguity (VUCA) [2]. In response, organizations are experimenting with emerging approaches to leadership development [3]. One recent trend is the aim of synthesizing the growing popularity of self-management [4,5] with the development needs of individuals [6] and the pursuit of agility [7]. As a result, participatory and more individualized approaches to leadership development gain ground [8]. A participatory approach involves the target group of leaders—the leaders who will participate in a development program—in the development and goal setting of the program together with their senior management. Conceiving a leadership development program together with its participants in a co-creation process helps to align hierarchical levels. It supports embedding leadership in the day-to-day business of the organization (cf. [9,10]).

The principal ambition is to create a work environment where both employees and leaders may flourish, self-develop, and meaningfully contribute [11]. New and emerging practices aim to contribute to both organizational performance and employee well-being [3]. However, studies on the effects on business performance are lacking [12]. The current study aims to fill this gap: The present study’s leadership development program was shaped in a co-creation process and sought to improve actual business performance, decrease sick-leave absenteeism, and elevate motivation.

### 1.1. Self-Determination Theory: A Psychological Theory Underpinning Leadership Development

We chose self-determination theory (SDT) as the theoretical point of departure for the current study because of its integrative approach. SDT links a theory of human development and motivation with a view on organizational effectiveness and leadership. Through basic psychological needs theory (BPNT) [13], and organismic integration theory (OIT) [14], SDT explains how employees take in extrinsic regulations (regulations external to the individual), such as purpose, mission, vision, goals, targets, procedures and controls, and integrate these regulations with their sense of self [15]. The integration process is an essential aspect of the dialectic view on human development in SDT [13]. Consequently, leadership and organizational experiences are experiences that are integrated into a person’s sense of self [16]. When, in a work context, extrinsic regulations are positively integrated into the sense of self, SDT speaks of autonomous motivation: individuals may identify with extrinsic regulations because they, for example, identify with the purpose of the department or organization and find personal meaning and value in the given purpose. Identifying with a purpose may add to a person’s sense of meaningfully contributing and opens the possibility of self-directing one’s motivational energy towards that purpose and supports well-being [11]. In answer to the question “why do you do this work”, a person may answer “because it is important to me” or “it aligns with my personal values” [17]. The more a person identifies with the organizational goals, values, purpose, or tasks at hand, the more a person is inclined to take on responsibilities and self-manage or self-lead [15,18].

In short, SDT posits that people flourish, are optimally motivated, perform well, and contribute positively to the organization’s performance by positively identifying with and positively integrating leadership experiences [11]. Hence, the current study’s leadership development program was designed to support leaders in learning how to create work contexts where the integration process of extrinsic regulation is positively facilitated.

### 1.2. Autonomy Satisfaction

Autonomous and intrinsic motivation associate with the satisfaction of the basic psychological needs of autonomy, competence, and relatedness [16]; the fulfilment of these needs induces optimal human functioning, autonomous motivation, and well-being. BPNT defines autonomy as the need to be the author of one’s own fate and to be involved and heard in matters of personal interest [13]. Relatedness refers to the need to be loved, held, and cared for, and to have meaningful relationships with others [19] and competence relates to the need to feel effective relative to one’s environment [20] and to be good at something. Ryan and Deci [11] claim a specific role for autonomy satisfaction in the system of needs. In many instances, the capacity to self-organize precedes the fulfilment of the need for competence and relatedness. Autonomy—in the sense of agency—is a necessary ingredient for individuals to initiate behaviors through which these other two needs may be realized [11]. Since the individual experience of autonomy is also a socio-contextual phenomenon [14], it very much coheres with the active, positive role of the leader and whether team members experience an elevated sense of autonomy, which may induce a better realization of the needs for competence and relatedness [21]. Leaders that fulfil the basic need of autonomy create a positive precondition for elevated work engagement [22], well-being [23], and performance [24].

### 1.3. Engaging Leadership and Positive Outcomes

The engaging leadership concept was selected because it draws on BPNT and aims explicitly to identify leadership behaviors that may induce work engagement through the satisfaction of basic psychological needs [25]. Through BPNT, engaging leadership grounds itself in human motivation theory [13]. It is a human-centered leadership approach, as opposed to more traditional leader-centered methods, such as transformational leadership [26], or authentic leadership [27]. Moreover, engaging leadership aspires to create the conditions that nurture performance [28] through the aspects of empowering, strengthening, and connecting [25]. Empowering corresponds to the need for autonomy and emphasizes that employees should be engaged in matters that concern them and should be made part of and have a say in the larger whole [16]. An engaging leader allows employees to craft their work and self-direct within a clear and structured context [18]. Strengthening fosters the need for competence through plentiful feedback—mostly positive feedback—and creating space for employees to personally and professionally grow and self-develop [13]. Connecting addresses the need for relatedness, through nurturing personal and meaningful relationships across hierarchical levels and setting a context of care and support [11,18]. Defining leadership behaviors to align with need satisfaction is a promising approach. A recent study found basic needs to mediate the relationship between engaging leadership and work engagement [29] and identified need fulfilment as a potential explanatory mechanism. Previous studies also found to basic needs to mediate between leadership and work engagement [30]. Because of its explanatory value need satisfaction is sometimes referred to as the unifying principle [31].

Prior studies on the effectiveness of leadership development within SDT refer to “autonomy-supportive leadership” and found that leadership behaviors promoting the satisfaction of basic needs and autonomous motivation can be learned [32]. Still, it seems that leaders participating in leadership development programs benefit more than team members [33,34]. Researchers found autonomy-supportive leadership to be effective in fostering better performance in domains such as sports coaching [35], educational and academic settings [36], and healthcare [37]. In the realm of organizational psychology, however, most studies measure subjective performance based on self-reports in terms of in-role or extra-role performance [38] or based on performance appraisals by supervising leaders [39]. We have found no studies within the domain of SDT reporting on actual business performance as an outcome measure. In the present study, we argue that co-creation by engaging team leaders (the participants) in the design and development of the intervention is an expression of autonomy support. We expect that this engagement will support the subsequent realization of business results and well-being.

### 1.4. Documenting Business Performance

Periodical progress in business performance is generally measured using the balanced scorecard methodology developed by Kaplan and Norton [40], which aims to align performance indicators with strategic business objectives and definitions of business excellence [41]. For supply chain management, an additional instrument, the supply chain operations reference (SCOR) model, has been developed. The model supports organizations to develop the ability to manage the full scope of the supply chain, from a functional level to inter-organizational integration [42]. In the current study, the aim was to test whether the leadership development initiative could positively contribute to actual business performance through carefully selecting a KPI (key performance indicator). A KPI is a metric indicating how a company or department is performing relative to a predefined key business objective. The selected KPI had to be a key indicator for the intervention group’s performance, and it was considered essential that team leaders and their team members could influence the KPI. For this study, representatives of the team leaders, their supervising manager and the global department head of the intervention group selected one core business KPI in the process of co-creation—orders booked on time (OBOT)—the percentage of the total number of new orders booked per month within the target parameter of 48 hours after receipt from the sales department. The target was to achieve 95% OBOT, and team members logged progress in the company’s CRM-software and reported monthly.

### 1.5. Absenteeism

Another widely used KPI is the level of sick-leave absenteeism, which is considered an objective, operationalized performance measure linked to organizational performance [43]. Positive leadership styles, such as transformational leadership, contribute to lower levels of absenteeism [44], whereas more controlling leadership styles [16] and passive-avoidant styles of leadership are associated with higher levels of absenteeism [45]. Additionally, increased autonomy satisfaction predicts lower levels of absenteeism [39,46]. Organizations with lower levels of absenteeism report higher engagement and productivity [47], while higher frequencies of sick-leave absenteeism are associated with poor performance [48]. Lower levels of sick-leave absenteeism lead to lower costs and, just like productivity gains and increased performance, can be used as valuable input for business metrics, such as a return-on-investment (ROI) calculus, which is one of the most widely used performance evaluation metrics in the management literature [12,49]. We argue that engaging leadership, with its base in self-determination theory and need satisfaction, will also contribute to lower absenteeism rates. However, there are no prior empirical studies that we can draw on that have specifically explored this relationship for engaging leadership.

For the present study we used individual sick leave reports documented by the HR department and the occupational health and safety provider. Two separate periods were analyzed to measure sick-leave absenteeism: a pre-intervention period covering the 12 months preceding the intervention and a post-intervention period covering the 12 months after the program [50]. The measures applied were sickness frequency, the absence rate, and the net number of lost workdays. Individual sickness frequency was expressed as the number of current and new sick-leave spells divided by the number of full-time equivalents (FTEs). The absence rate was defined as the number of lost workdays per employee reported sick, divided by the total number of available workdays times one hundred. The net amount of lost workdays consisted of the reported workdays per month lost as a result of absenteeism.

### 1.6. The Current Study

The primary aim of the current study was to improve objective business performance and decrease absenteeism through a leadership development program of which co-creation formed an integral part to support autonomy of participating midlevel team leaders. The secondary aim was to test whether the content and the design of the leadership program would benefit participating team leaders and whether the positive effects would spill over to the team members. Three hypotheses were formulated:

**Hypothesis** **1.**
*As a result of the leadership program, (a) the business performance of the department of the intervention group will increase to or exceed the agreed target level, and (b) absenteeism within the department will decrease.*


**Hypothesis** **2.**
*(a) The team leaders participating in the leadership development program will report higher levels of engaging leadership compared to the control group, and (b) they will experience an increase in their levels of autonomy satisfaction and (c) intrinsic motivation.*


**Hypothesis** **3.**
*The effects of the leadership program will positively spill over to the team members, who are expected to benefit from the program indirectly. As a result, team members will perceive more engaging leadership and higher levels of autonomy and intrinsic motivation than those whose leaders did not participate.*


## 2. Materials and Methods 

### 2.1. Participants and Procedure

The present study used a quasi-experimental, pre-test–post-test control group design. The intervention and control groups were selected from the same multinational organization, which develops, produces, and services health systems. The intervention and the control group were responsible for back-office processes and delivered their information and services to other departments of the organization. The intervention group consisted of team leaders and team members of the customer fulfilment centre, whose main task is to manage the back-office processing of new orders from sales to production, shipping, delivery, and invoicing. The control group members were information analysts from the information management department, whose jobs are to gather, analyze, control, and distribute information across the organization. The control group was selected on comparable qualities to the intervention group, such as department size in full-time equivalents, business process (back-office, administrative, Lean management, service role to other departments), and locality. The intervention group consisted of 14 team leaders (*N* = 14, female 29%) and 148 team members, and team size averaged 10–12 members per team. At pre-test (T0), 13 team leaders, 93% response rate, female = 29%, age = 38.57 (standard deviation, SD = 8.58), tenure 3.88 (SD = 2.35) and 106 team members, 71% response rate, female = 51%, age = 39.32 (SD = 9.55), tenure 2.5 (SD = 2.23) completed the survey. At follow up (T1) eight months later, 14 team leaders completed the questionnaire, 12 of whom had also responded at T0; this difference in follow up was due to a staff change in the team, as one of the team leaders left the department. Regarding the team members, 109 responses were received at T1 (74% response rate), 81 respondents had also completed the survey at T0.

Unfortunately, the survey protocol of the organization forbade to link team members with their leaders in teams with fewer than 15 members for reasons of anonymity, implying that the team leaders and team members had to remain separate groups and could not be nested. As a consequence, within both the intervention and control group, two separate groups were included: (1) team leaders and (2) team members, instead of one leader per team with his or her direct reports. The control group consisted of 52 team leaders and 218 team members. At T0, 39 team leaders, 74% response rate, female = 30%, age = 42.42 (SD = 8.38), tenure 2.06 (SD = 1.10) and 119 team members, 55% response rate, female = 43%, age = 44.36 (SD = 11.70), tenure 1.95 (SD = 1.31), completed the survey. At T1 eight months later, 23 team leaders responded, 21 of whom had also participated at T0. A total of 104 responses were received from the team members, of whom 62 had also completed the survey at T0.

### 2.2. The Leadership Intervention

The leadership development program consisted of three phases: (a) co-creation; (b), intervention or execution; and (c) evaluation and sustainment [10,51]. Figure 1 schematically displays the structure and content of the program. The initiation and co-creation phase of the program took two months. The structure, operational details, and parameters for success were discussed with the team leaders and their supervising manager, the senior leadership team, and the global department head, who also initiated the program. Actively involving the team leaders in the design phase of the project mirrored the preferred leadership style that the department sought, which was reflected in the department’s slogan for the project: “Co-creating a great place to work.” Consequently, co-creation continued during the months of the intervention. The trainers remained in close contact with the global department head, his team, the supervising manager, and the participating team leaders. The team members were not involved in co-creation nor would they participate in the leadership development program’s training sessions. Team members were asked to complete an online survey, to which their team leaders personally invited them. The program was launched with the pre-test survey to establish a baseline and unfolded eight months, after which the T1 follow-up survey was administered.

With the control group, the design and purpose of the program were discussed with representatives of the control group’s senior leadership team, who then informed the team leaders, who invited direct reporting team members to participate in the survey on a voluntary basis through email. Team members were, however, not informed about the intervention group’s leadership program.

The second phase of the intervention consisted of three building blocks (Figure 1): six training days with a 6–8-week interval, three peer-consultation sessions between the training sessions, and two one-on-one coaching sessions. The design was modelled on the professional experience of the trainers, on previous intervention studies within SDT, e.g., [33], and studies conducted by the Center for Creative Leadership [51].

The training program (Figure 1) commenced with a two-day off-site session for the team leaders and their supervising manager. Day one opened with a restatement of the purpose of the development program and a welcome speech by the department’s global head and focused on improving team relations and trust through feedback and exercises. Day two explained the engaging leadership concept and participants formulated shared goals for the program. The third training day (six weeks later) focused on the progress made in generating positive outcomes relative to the goals and ambitions from the first training session. Furthermore, the findings of the baseline measurement were presented, and exercises and practice in personal resilience were offered. Coaching leadership and coaching conversation exercises formed the core of day four. Day five explained the SDT theory of motivation (motivation continuum and basic psychological needs) [13,16] and offered exercises in motivational coaching and handling push back and negative emotions. On day six the results realized were evaluated, and the progress made was celebrated.

Three guided peer-consultation sessions were offered between the training days. Peer-consultation provides a structured method of facilitated conversation for improving personal functioning and professional performance, addressing leadership challenges, and engaging in group problem solving based on mutual support and consultation with peers. The team leaders also had two personal coaching sessions to address their leadership challenges confidentially. The peer consultation sessions and coaching sessions were facilitated by two coaches, who worked according to the International Coach Federation (ICF) guidelines.

### 2.3. Instruments

Business performance was measured with the metric OBOT as defined earlier. The business metric was selected because it was considered the essential business metric for the department. Absenteeism was measured following the procedure described previously. The company’s overall target for absenteeism was to remain below a 3% threshold. At the time of the initiation phase of the project, absenteeism in the department had risen to 7.7%, which raised a red flag to senior management and accelerated the development of the leadership development program. 

Engaging leadership (EL) was measured with the 9-item Engaging Leadership scale developed by Schaufeli [25], which was distributed in two versions: a self-assessment for team leaders and an instrument for team members to assess their team leader. The aspects of strengthening, connecting, and empowering were each represented by three items. The following is an example item for team leaders (strengthening): “At work, I encourage team members to develop their talents as much as possible.”; for team members this is: “At work, my supervisor encourages team members to develop their talents as much as possible.”; an example of connecting is: “I encourage collaboration among team members.”; for team members this was: “my supervisor encourages collaboration among team members.”; empowering was represented through “I encourage team members to give their own opinion”; for team leaders, and for team members this item was introduced with “my supervisor encourages (…).”. Responses were measured on a 5-point Likert scale ranging from 1 (completely disagree) to 5 (completely agree). The values of the Cronbach’s alpha indicator for reliability are shown as follows: α = 0.92 (T0) and α = 0.93 (T1) for team members and α = 0.82 (T0) and α = 0.78 (T1) for team leaders.

Autonomy satisfaction (AS) was measured with the three items from the Balanced Measure of Psychological Needs (BMPN) Scale developed and validated by Sheldon and Hilpert [52]. The items were measured on a 5-point Likert scale (1, completely disagree; 5, completely agree). For example: “I am really doing what interests me.” The reliability values were: α = 0.70 (T0) and α = 0.66 (T1).

Intrinsic motivation (IM) was measured with the three items for intrinsic regulation from the Multidimensional Work Motivation Scale [17]. The items were scored on a 5-point Likert scale (1, completely disagree; 5, completely agree). For example: “Because I have fun doing my job.” The reliability values were: α = 0.83 (T0) and α = 0.86 (T1). See Appendix A for the survey items.

The business outcome measures were reported every month to the global department head and continued after the intervention had ended. In contrast, the survey was administered at two time-points, at pre-test and post-test after the intervention program had completed. The business measures were followed for another six months beyond the intervention program to evaluate whether the intervention results were sustained into the next year.

## 3. Results

### 3.1. Preliminary Analyses

First, we screened the data for multivariate outliers using the Mahalanobis distance procedure, which identified one outlier in the intervention group at pretest: the respondent had dropped out and was thus not considered in the analyses. Skewness and kurtosis were verified to be within an acceptable range. The MCAR (missing completely at random) procedure was run [53,54] to establish if data were missing completely at random. The test returned that data were missing completely at random: χ^2^(103) = 119.01, *p* = 0.134. Nevertheless, it was checked whether one of the variables could be predictive of subsequent dropout of respondents through examining systematic attrition [53] for the separate groups (Table 1). One team leader in the intervention group with a very low score on engaging leadership (presumably the outlier mentioned above) dropped out, which probably was related to the staff change mentioned earlier—it was assumed this was the team leader that had left the department. For team members in the intervention group, low scores on engaging leadership seemed predictive of medium risk of later dropout (*t*(104) = 3.29; *p* < 0.01; 95% CI, confidence interval, [−0.91, −0.23]; *d* = 0.69). The other two variables (autonomy satisfaction and intrinsic motivation) did not and presumably counterbalanced the dropout risk. By the same token, low scores on autonomy satisfaction may have contributed to a medium-low risk of team members dropping out in the control group (*t*(117) = 2.23; *p* < 0.05; 95% CI, [−0.53, −0.31]; *d* = 0.41). No further steps were taken since the missing data were MCAR.

The descriptive statistics were calculated for both the intervention and control group at baseline (Table 2). At pre-test the team leaders in the intervention group rated themselves higher on engaging leadership than their direct reporting employees assessed them *t*(117) = 1.92, *p* = 0.06, 95% CI [−0.01, 0.81], *d* = 0.63. Although the difference was nonsignificant, the effect size was of medium strength. In the control group the difference between the leaders self-assessment and their team members was similar in strength and significance: *t*(156) = 3.00, *p* = 0.003, 95% CI [0.12, 0.56], *d* = 0.62.

The means, standard deviations, and intercorrelations between variables were examined at the two time-points for the team leaders and the team members (Table 3).

### 3.2. Business Outcomes for KPI Performance and Absenteeism (Hypothesis 1)

Hypothesis 1a predicted that business performance would increase to or exceed the target level set. To assess the department’s business performance, as measured by OBOT, the “N-1” chi-square test [55] was performed. The number of orders booked per month differed over the year. Peak moments were found at the end of each quarter and the two months at year-end. To compare the results, we have looked at business outcomes in March (quarter-end and pre-test), November (year-end and comparable with March and post-test), and in June the following year (quarter-end). December is by far the busiest month and not comparable with any other month, hence November was assessed. OBOT at T0 (March) was 87% of 22,368 orders, compared to 92% of 22,165 orders at T1 (November), which is a significant increase: χ^2^ (1, *N* = 22,165) = 295.88, *p* < 0.001, 95% CI [4.43, 5.57]. The percentage increased another 3% in the six-month period after the intervention (June) to 95%: χ^2^ (1, *N* = 22,929) = 167.50, *p* < 0.001, 95% CI [2.55, 3.46]. Additionally, the average number of orders booked on time per fulltime equivalent developed positively: At pre-test, this was 304 OBOT/FTE, at post-test, this was 324, and 6 months after post-test this was 357.

Hypothesis 1b predicted that absenteeism decreases as a result of the leadership program. The sick-leave absenteeism data obtained from the service provider were analyzed using nonparametric statistics due to the skewed nature of the sickness absence data. We applied the Wilcoxon signed ranks procedure for related samples to test the *within*-group differences between the pre- and post-test measurements. As depicted in Table 4, the sickness frequency dropped but not significantly (*p >* 0.05). In contrast, the intervention group’s sick-leave absence decreased significantly over the twelve months after the intervention by 3.4% (*p* < 0.001). The total net number of lost workdays per month declined by almost 60%, from an average of 113 per month in the twelve months preceding the leadership program to 66 after the program (*p* = 0.02). Overall, the outcomes confirmed the hypothesis that sick-leave absenteeism would drop as a result of the leadership intervention program.

The “between”-group differences between the intervention and control groups were tested with the Mann-Whitney U test for independent samples. The pre-test outcomes comparing the intervention and control groups indicated significant differences (*p* < 0.001) between the two groups for all absenteeism tests performed, except for frequency (Z = −2.72, *p* = 0.007). However, the differences between the two groups were non-significant in the post-test analysis, except for frequency (Z = −2.60, *p* = 0.009), presumably due to the significant decreases in the intervention group and the parallel increases in the control group regressing to the mean.

### 3.3. Effects on Team Leaders and Team Members (Hypotheses 2 and 3)

Hypothesis 2a predicted that team leaders would (a) report higher levels of engaging leadership (EL) and (b) display an increase in autonomy satisfaction (AS) and (c) intrinsic motivation (IM). A multivariate test of variance was used to test the differences between the two groups over the two time-points, with EL, AS, and IM at T1 as dependent variables. The same variables were used as covariates at pre-test, to adjust the means between the intervention and control groups at the pre-intervention level. Additional checks on the underlying assumptions for a successful two-way multivariate analysis were conducted. The regression slopes were linear, and homogeneity checks were performed. Levene’s test to assess the equality of the variances confirmed there were no significant differences in variance between the two groups or the leads and members. Box’s M test was conducted to assess the equality of co-variance and supported the assumption that the co-variance matrices between the groups could be assumed to be equal. As far as demographics variables are concerned, age had a positive and tenure had a negative effect on engaging leadership and motivation for team leads. Conversely, tenure had a positive effect on engaging leadership, autonomy, and motivation for team members. Therefore, we decided to control for age and tenure for team leads and tenure for team members.

Two separate analyses, one for team leaders and one for team members, were conducted. The intervention effects were non-significant for the team leaders and team members (Table 5). Considering the effects of small sample size on potential significance, we continued to test whether the effect sizes were relevant in magnitude. The partial epsilon squared is said to be the least biased effect measure for small samples [56]. It is interpreted using Cohen’s [57] rules of thumb: 0.01, 0.06, 0.14 for small, medium and large effect sizes respectively. We calculated the partial epsilon squared values with the formula by Albers and Lakens [58] and found no effects for engaging leadership. In contrast, we found a very large effect (ε_p_^2^ = 0.25) for autonomy satisfaction and a large affect for intrinsic motivation (ε_p_^2^ = 0.12). Hence Hypotheses 2b and 2c were supported by the data, whereas Hypothesis 2a was not.

Hypothesis 3 predicted that the effects of the leadership program would positively spill over to team members. However, no relevant changes were found in the team members’ levels of perceived EL, AS, or IM (Table 5). Next, the effects within the intervention group were assessed. The analysis indicated a significant (*F*(1, 88) = 4.23, *p* = 0.04, ε_p_^2^ = 0.04) increase in AS for team leaders (M = 3.83, SD = 0.48) compared with team members (M = 3.54, SD = 0.62). Also, the IM scores for team leaders (M = 4.06, SD = 0.49) were significantly higher (*F*(1, 88) = 5.88, *p* = 0.02, ε_p_^2^ = 0.05) than those of team members (M = 3.68, SD = 0.69). The corresponding effect sizes indicated a medium-small (AS) to medium (IM) strength.

## 4. Discussion

The present intervention study aimed to generate positive business outcomes in terms of KPI performance, reduced absenteeism, and elevated motivation and well-being through an engaging leadership development program. The program consisted of an initial co-creation phase. It targeted midlevel team leaders who were subsequently trained in engaging leadership and the satisfaction of basic psychological needs through six one-day training sessions. Peer consultation and personal coaching were offered to support the integration of the program contents in small groups and one-on-one coaching sessions. The program resulted in a significant increase in KPI performance (OBOT), which continued after the program had ended and was in support of Hypothesis 1a. The total rise in OBOT had a beneficial monetary impact (8% more orders booked on time per month with the same number of FTEs and resources). The department was also able to invoice earlier, which led to a substantial positive effect on cash, as the department head reported.

Sick-leave absenteeism among the department’s team leaders and team members decreased considerably during and after the program, which outcome was as expected (Hypothesis 1b). The longer-term sick-leave absenteeism analysis, averaging the 12 months before and the 12 months after the intervention, resulted in a significant drop in absenteeism to 2% in the intervention group. Additionally, the net number of lost workdays dropped by almost 60%. As with the KPI performance, the sick-leave absence rate continued to improve post-program, underscoring the sustainability of the intervention effects over time and beyond the program. With these two outcomes, KPI performance and absenteeism, the department’s leadership considered the intervention to be a success.

Parallel to the organization’s metrics, the team leaders and team members’ psychological effects were measured and referenced with a control group. As expected, and in support of Hypotheses 2b-c, the team leaders displayed higher levels (very large effect size) of autonomy (2b), and motivation (2c, large effect size) relative to the control group. The effect sizes match the general conclusions from a meta-analysis of leadership development programs in the private sector (between 1952 and 2002) based on similar study designs [59,60]. For engaging leadership, however (Hypothesis 2a), no effect was registered.

The current leadership development program was titled “Co-creating a great place to work” and consisted of three phases: co-creation, intervention, and evaluation and sustainment. Co-creation was considered an expression of autonomy-supportive and engaging leadership. It aimed to facilitate productive dialogue between the leadership levels and helped align the global leadership team and the local team leaders. The selected business KPI, the ambition to lower sick-leave absence, and the aim to increase the levels of motivation via engaging leadership and autonomy satisfaction all stemmed from this first phase. Presumably, the co-creation process, as an integral part of the development program, facilitated achieving the observed changes in the autonomy and motivation of the team leaders relative to the control group.

Despite the positive effects of autonomy and motivation, the team members reported no relevant changes, which was unexpected (Hypothesis 3). The ambition to indirectly improve employee well-being by training their team leaders may prove problematic [61]. Other leadership intervention studies also found the intervention effects to be more substantial for the participating leaders than the indirect, spill-over effects for employees [33,34].

### 4.1. A Lesson Learned and a Suggestion for Future Programs

Team members were not involved in co-creation and did not participate in the program, which may explain the absence of effects of the program for this group. A recent longitudinal study, however, illustrated the beneficial impact of perceived line management support and active employee participation on job outcomes [10]. Involving employees in the design and execution of leadership development might have contributed to the program’s overall effectiveness. Additionally, actively engaging team members in co-creation corresponds with the growing popularity of participative and self-management approaches, replacing traditional command-and-control hierarchies [4,62]. Additionally, it aligns with the conceptualization of SDT [13] on shaping autonomy-supportive work environments where employees may flourish.

A practical proposal to involve employees as active participants may consist of including approaches such as action learning, where team leaders and team members collectively work on joint assignments [63]. Including team members, or employees more broadly speaking, may induce further positive changes in engagement and performance beyond the current study. Since it is broadly endorsed that employee engagement links to positive business outcomes, including financial performance [64], improved effectiveness of a comparable future leadership intervention should be expected.

### 4.2. Contribution to Knowledge Development

The present study is the first in the domain of SDT to report the impact of a leadership development intervention on actual business performance, as measured with business key performance indicators (i.e., productivity and absenteeism). Previous studies published on leadership impact within SDT reported on, for example, performance evaluations by supervising managers [39] or self-reported in-role versus extra-role behaviors [38]. Another study measured the impact of autonomy-supportive leadership on employee perceptions [34], and Deci et al. [33] reported on the positive effects of a leadership intervention on supervising managers’ orientations. Similar to the present study also in Deci et al.’s research, the impact on employees was less conclusive. The present study’s contribution to leadership and SDT is relevant because it adds to the range of beneficial results that may be obtained through engaging leadership behaviors (connecting, empowering, strengthening) [25] aiming to satisfy basic psychological needs [13]. It also substantiates the claim that basic need satisfaction through autonomy-supportive leadership behaviors, such as described in engaging leadership, positively associates with enhanced performance [11,65,66].

Moreover, the present study also contributes to leadership development. The design of the leadership intervention facilitated the collective leadership experience through the participatory process of co-creation of senior leaders and the participating team leaders, and hence followed the call of Petri [67] for more collective, networked approaches to leadership development. Engaging the participating team leaders supported the inclusion of the selected business KPI, absenteeism, and motivation as pillars of the progam. Furthermore, engaging leadership behaviors [25] that were taught in the training sessions followed the principles of coaching for leadership development, as described by Ting [68]. Examples are: creating a safe and challenging environment (connecting), choosing facilitative leadership over directive action planning (empowering), and promoting learning through collectively evaluating experiences and applying these to improve the business outcomes further (strengthening).

We believe that designing a leadership development program in a real-time work environment with a real business challenge (cf. [69]) answers to the need for more contextually embedded leadership initiatives [70] and measurable outcomes [71]. We suggest that future leadership development practice can benefit from the incorporation of leadership development programs—like the one in the current study—in the day-to-day processes of departments and organizations [3].

### 4.3. Implications for Practitioners

There are a few important implications of the current leadership development program for practitioners. First, involving the target group of participants in a co-creation process is beneficial. It helps create direction, alignment, and commitment (DAC) [72] across two or more hierarchical levels. Additionally, it answers to the need to work more peer-like and collaboratively in an autonomy-supportive manner. The second implication is to define program objectives that enthuse and inspire the targeted leaders and senior management. Improving business results should be one of the purposes and be positioned within the organization’s broader goals to provide relevance for both participants and the organization. Thirdly, it is essential to use the program as a lever to embed autonomy-supportive engaging leadership in everyday practice. The leadership development program is more likely to become an actual practice when it is situated in a real-time context with real consequences. The fourth implication is to extend the program to, at least to a certain extent, other employees, not just team leaders or managers. Lastly, it is recommended to blend classroom training, with additional instruments such as peer-consultation in small groups, one-on-one coaching, and action learning.

### 4.4. Limitations

A major strength of the current study is that it was carried out in a real-world and, thus, dynamic organizational context. Despite its high ecological validity, it has a few limitations. The study is based on a single intervention within one organization. A second limitation is that the study relies on two measurement points, where three or more measurements over time could have added to the solidity of the findings, which is supported by the organizational outcomes displaying continued improvements six months after the second measurement. Third, the study incorporated small groups of team leaders, and the results could not be nested because of the survey protocol of the organization. Because of the small sample size, the power was too low to obtain significant parameter estimates and we relied on effect sizes only. We selected the partial epsilon squared because it is said to be the least biased in small samples [58]. A future intervention study would benefit from larger groups of team leaders, which makes significance and robustness of relevant differences more feasible. Fourth, the quasi-experimental design plus the co-creation phase of the program, in which the senior leadership and team leaders participated, may have drip-fed information on the proposed leadership intervention to team members previous to the initial survey. Hence, it cannot be assumed that the intervention group’s team members were completely unaware of the experiment when filling out the survey. This limitation did, however, not extend to the control group because only its head of department knew about the experiment in the intervention group.

Consequently, we could not distinguish the leadership impact of the various team leaders on their teams. Most likely, some team leaders had a positive and significant impact on their team members’ AS and IM, which was learned from the peer group consultation sessions and the individual coaching sessions with the team leaders during the program. However, for reasons of confidentiality, these sessions were not recorded or transcribed. Additionally, the small sample size for team leaders and the resulting low statistical power brings the risk of Type II errors for the interpretation of the psychometric results. A fifth limitation is the quasi-experimental study design itself, which makes it more challenging to conclude the causality effects of the experiment. To balance this limitation, some measures were taken, such as the selection of a comparable control group and the correction for mean differences to create equivalence at baseline [73]. Another limitation on causality is the potential impact of unobserved managerial and organizational effects on observed performance, which may explain additional the variance in the outcome variables [74]. We could not control for such influences.

A seventh limitation is the substantial difference in sick-leave absenteeism between the intervention and control group at pre-test. During the intervention period, both absence rates seemed to regress towards the mean, which potentially implies that it may be a natural regression rather than a real effect [75]. Nevertheless, we stand by the claim that it was an intervention effect for two reasons. Firstly, because after a subsequent reorganization, two years later, the absence levels in the intervention group were back at their high pre-intervention levels. Secondly, because the drop in the intervention group was significant compared to the absenteeism within the organization at large, and even dipped below the organization’s 3% threshold.

## 5. Conclusions

The present study showed that a leadership development program focusing on engaging leadership and psychological well-being led to significant positive business results and lower absenteeism. Additionally, the intervention had a moderate to large effect for the motivation of team leaders and a substantial impact on the experienced autonomy of team leaders compared to the control group. The business outcomes exceeded the leadership team’s ambitions and continued to improve post-program. The predicted positive spill-over effects on team members were absent despite the positive psychological effects on autonomy and motivation among team leaders and their subsequent well-being.

From a practical point of view, co-creating a leadership program by actively involving its participants helped create direction, alignment, and commitment. It also supported the realization of business outcomes in a spirit of collaboration and continuous improvement. Embedding leadership development in everyday business processes with concrete business objectives and extending co-creation throughout the program supports team leaders in developing engaging leadership in their day-to-day practice and contributes to tangible results. The next step is to find ways to extend the program and engage team members in co-creation and implementation. Considering the current trend of self-management and agility in business organizations, we argue that leadership development should not be limited to management or a select group of high-potential individuals.

Despite the positive results of the program, generalization across organizations is limited, because it comprised a single intervention in one organization and a relatively small group of team leaders. Additionally, co-creation may lead to different foci in different organizations and influence the outcomes. However, we believe that the described approach can be strengthened through replication in other organizations. The present study contributes to the practical knowledge of the relatively new concept of engaging leadership and its successful application in leadership development programs.

## Figures and Tables

**Figure 1 ijerph-17-04515-f001:**
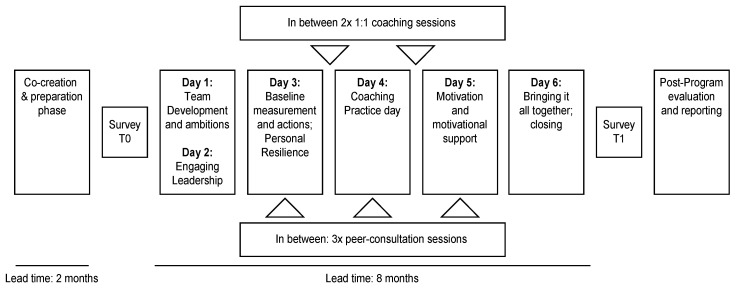
This Figure displays the program’s structure and content, starting with a two-month preparation and co-creation phase in which the goals and parameters for success were defined between the global leadership team of the department and the local leadership team that would participate in the program. Then, the pre-test survey was sent out to establish baseline (T0). The training intervention consisted of 6 training days with a 6–8-week interval. The first training session lasted two consecutive days. In between, three peer consultation sessions and two one-on-one coaching sessions were offered. The total duration was eight months, after which the post-test measurement (T1) was conducted. After the program had ended, business measures continued to measure progress.

**Table 1 ijerph-17-04515-t001:** Systematic attrition for engaging leadership, autonomy satisfaction and intrinsic motivation.

Intervention Group	Drop-Out				Difference				
	Yes		No						
	M	SD	M	SD	*t*	*df*	*p*	95% CI	*Cohen’s d*
*Leads*	*n*^a^ = 1		*n*^b^ = 12						
Engaging Leadership	1.75	0.00	3.79	0.45	NA				
Autonomy Satisfaction	4.00	0.00	3.83	0.48	NA				
Intrinsic Motivation	4.33	0.00	4.06	0.49	NA				
*Members*	*n*^a^ = 25		*n*^b^ = 81						
Engaging Leadership	3.20	0.95	3.77	0.69	3.29	104	0.01	[−0.92, −0.23]	0.69
Autonomy Satisfaction	3.37	0.80	3.54	0.62	1.12		0.27	[−0.47, 0.13]	0.23
Intrinsic Motivation	3.51	0.75	3.68	0.69	1.06		2.94	[−0.49, 0.15]	0.24
**Control group**									
*Leads*	*n*^a^ = 18		*n*^b^ = 21						
Engaging Leadership	3.92	0.69	3.86	0.64	0.28	37	0.78	[−0.37, 0.49]	−0.08
Autonomy Satisfaction	3.83	0.74	3.78	0.70	0.22		0.83	[−0.42, 0.52]	−0.07
Intrinsic Motivation	4.02	0.55	4.02	0.62	0.00		1.00	[−0.38, 0.38]	−0.00
*Members*	*n*^a^ = 57		*n*^b^ = 62						
Engaging Leadership	3.85	0.64	3.96	0.75	0.86	117	0.39	[−0.36, 0.14]	0.16
Autonomy Satisfaction	3.45	0.71	3.73	0.66	2.23		0.03	[−0.53, −0.31]	0.41
Intrinsic Motivation	3.69	0.70	3.84	0.64	1.22		0.22	[−0.93, 0.09]	0.22

*N* = 176; M, Mean; SD, Standard Deviation; *t*, t-statistic; *df*, degrees of freedom; *p*, significance (two-tailed); 95% CI, 95% Confidence Interval; ^a^
*n* refers to the number of drop-outs at T1. Additionally, there were 72 drop-ins: respondents that did not participate at pre-test but that did complete at post-test. They have been left out of the drop-out analysis; ^b^ here *n* refers to the total number of the same respondents who completed the survey at both time-points, hence *n* = 176.

**Table 2 ijerph-17-04515-t002:** Means (M) and standard deviations (SD) at baseline.

	**Intervention Group**			
	**Leads (*n* = 13)**		**Members (*n* = 106)**	
	**M**	**SD**	**M**	**SD**
Team Leads Self-assessment	4.06	0.51		
Engaging Leadership	3.40	0.81	3.66	0.73
Autonomy Satisfaction	3.59	0.70	3.53	0.65
Intrinsic Motivation	3.74	0.90	3.66	0.69
	**Control Group**			
	**Leads (*n* = 39)**		**Members (*n* = 119)**	
	**M**	**SD**	**M**	**SD**
Team Leads Self-assessment	4.33	0.39		
Engaging Leadership	4.03	0.61	3.99	0.67
Autonomy Satisfaction	3.75	0.72	3.58	0.68
Intrinsic Motivation	3.97	0.58	3.74	0.70

**Table 3 ijerph-17-04515-t003:** Means (M), Standard Deviations (SD), Intercorrelations (r).

	Team Leaders		EL (T0)	EL (T1)	AS (T0)	AS (T1)	IM (T0)	IM (T1)	Team Members	
	M	SD							M	SD
EL (T0) ^a^	3.87	0.71	1	0.70 ***	0.50 ***	0.46 ***	0.38 ***	0.37 ***	3.83	0.73
EL (T1) ^b^	3.86	0.55	0.50 **	1	0.38 *	0.41 ***	0.21 *	0.34 ***	3.89	0.77
AS (T0) ^a^	3.71	0.71	0.27 *	0.23	1	0.61 ***	0.60 ***	0.57 ***	3.56	0.67
AS (T1) ^b^	3.83	0.62	0.27	0.48 **	0.50 **	1	0.51 ***	0.66 ***	3.62	0.68
IM (T0) ^a^	3.92	0.67	0.22	0.12	0.77 ***	0.06	1	0.64 ***	3.70	0.70
IM (T1) ^b^	4.05	0.56	0.24	0.43 **	0.51 **	0.67 ***	0.43 *	1	3.73	0.72

^a^*n* = 52 (leads), 225 (members); ^b^
*n* = 37 (leads), 212 (members); * *p* < 0.05; ** *p* < 0.01; *** *p* < 0.001; M, Mean; SD, Standard Deviation; Below the diagonal: team leaders; above the diagonal: team members. Separate correlation matrices for the intervention and control groups are available upon request by the first author. EL, Engaging Leadership; AS, Autonomy Satisfaction; IM, Intrinsic Motivation; (T0), pretest; (T1), post-test.

**Table 4 ijerph-17-04515-t004:** Absenteeism in the intervention and control group.

	Pretest		Posttest		Wilcoxon ^a^		Mann-Whitney U
	M	SD	M	SD	Z	*p*	*p*
*Frequency*							
Intervention	1.59	0.84	1.29	0.51	−0.86	0.39	0.01
Control	0.62	0.47	0.78	0.40	−0.71	0.48	
*Absence rate*							
Intervention	5.40	2.49	2.03	0.74	−3.07	0.00	0.64
Control	1.03	0.20	1.96	0.72	−2.67	0.01	
*Net number of lost workdays*							
Intervention	112.58	54.40	66.46	29.62	−2.43	0.02	0.38
Control	5.44	7.47	52.72	29.97	−2.98	0.00	

*Note*. ^a^ Wilcoxon Signed Ranks Test; M, Mean; SD, Standard Deviation; Z, Z-score; *p*, significance (two-tailed).

**Table 5 ijerph-17-04515-t005:** Difference of intervention effects between intervention and control group.

		*F*	*p*-Value	η_p_^2^	ε_p_^2^
Leads	Engaging Leadership	0.02	0.91	0.00	−0.12
	Autonomy Satisfaction	3.94	0.08	0.33	0.25
	Intrinsic Motivation	2.23	0.17	0.22	0.12
Members	Engaging Leadership	0.11	0.74	0.00	−0.01
	Autonomy Satisfaction	0.97	0.33	0.01	0.00
	Intrinsic Motivation	0.89	0.35	0.01	0.00

*Note. F*, F-value; *p*-value, significance (two-tailed); η_p_^2^, partial eta squared; ε_p_^2^, partial epsilon squared; the test controlled for age and tenure for team leads and tenure for team members.

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
