# Peer review of "Business Results and Well-Being: An Engaging Leadership Intervention Study"

_ijerph, 2020, doi:10.3390/ijerph17124515_

Round 1

Reviewer 1 Report

Review manuscript titled: “Business Results and Well-being: an Engaging Leadership Intervention
Study”

  1. The paper is innovative, clearly written and deals with an important subject matter. Below, I put forward some comments.
  2. It is not clear if there was a social desirability effect originating from the control group knowing about the intervention
  3. Although there are practical considerations, please comment on why certain groups were selected for the intervention and not others.
  4. Constructs with only three items are just identified. Adding one or two will perhaps better represent a given construct. This is despite the generally acceptable alphas obtained.
  5. If certain performance levels went back to the ex-ante situation and there were no team members’ effects, then, there was no leadership effect (see line 510). This militates against your statements about the sustainability of effects after the intervention.
  6. One key element in any change is the presence, or absence, of reinforcing mechanisms (e.g., systems, processes, incentives) that should be in line with the desired new behaviors. The paper does not discuss this important component.

Author Response

RV (Reviewer comment) The paper is innovative, clearly written and deals with an important subject matter. Below, I put forward some comments.

RV It is not clear if there was a social desirability effect originating from the control group knowing about the intervention.

AW (Answer): Participants in the control group did not know about the intervention except for their department head, so we feel safe to assume there was no social desirability effect.

Text added at line 232 (Track Changes off): “Team members were, however, not informed about the intervention group’s leadership programme.”

RV: Although there are practical considerations, please comment on why certain groups were selected for the intervention and not others

AW: The researchers were invited by the senior leadership team of the intervention group because of their challenges with sick leave absenteeism. Additionally, the senior leadership believed an alternative approach to leadership could support both performance and lower absenteeism rates. This combination of things triggered the start of the project. The control group was selected later because we aimed for a quasi-experimental pretest-posttest control group design to measure whether the program actually had impact. We looked for a department within the same organization resembling the intervention group in size and business process. For example: both groups were back-office departments in service of other departments of the organization.

Text added at line 192 (Track Changes off): “The control group was selected on comparable qualities to the intervention group, such as department size in full-time equivalents, business process (back-office, administrative, LEAN management, service role to other departments), and locality.

RV: Constructs with only three items are just identified. Adding one or two will perhaps better represent a given construct. This is despite the generally acceptable alphas obtained.

AW: Engaging leadership was represented through a 9-item scale. For both autonomous motivation and intrinsic motivation we only had three items each. Hence, we do not have extra items we may add. Intrinsic motivation came from the Multidimensional Motivation at Work Scale (MMWS) of Gagné et al. (2014) and autonomous motivation from the Balanced Measure of Psychological Needs (BMPN), Sheldon and Hilpert (2012).

Additionally, we would like to point to recent literature indicating three-item versions of psychometric scales do yield the same results as more extensive ones. Three recent examples: Kjell and Diener, 2020; Schaufeli et al., 2019; Rahmadani et al. 2020.

RV: If certain performance levels went back to the ex-ante situation and there were no team members’ effects, then, there was no leadership effect (see line 510). This militates against your statements about the sustainability of effects after the intervention. | disagree, explain

AW: The results obtained throughout the leadership intervention on both the business KPI and on absenteeism continued to improve over a period of six months after the posttest survey.

These business performance and absenteeism data were made available through the organization, who, of course, continued to measure the departments performance.

Two years after the intervention the international board of management announced to offshore the intervention group’s department. This caused unrest among employees. As a result, performance dropped and absenteeism increased again. This is the “reorganization” referred to in line 510 (Track Changes off). Please note that in the newly uploaded document line 510 now is line 556 when ‘track changes function is “off.”

Until that announcement the results of the department remained consistent with the intervention, and, as reported, continued to improve.

Hopefully this resolves the seeming contradiction.

RV: One key element in any change is the presence, or absence, of reinforcing mechanisms (e.g., systems, processes, incentives) that should be in line with the desired new behaviors. The paper does not discuss this important component.

AW: In the training sessions, team leaders were invited to apply the process of co-creation in their own teams and to use approaches such as peer-consultation to involve their teams into continuous improvement conversations. Teams discussed weekly what worked well for them and what needed improvement. Each week suggested improvements were made actionable and progress was assessed the next week. The formal process, besides specific supply chain management procedures of the organization, was LEAN management, of which the continuous improvement process was an aspect.

Up until the leadership intervention there was considerable opposition to LEAN from team members and team leaders, because they felt it was forced on them by senior management in a very controlling manner, as we learned from interviews that were conducted in the orientation phase of the programme. This phase was not described in the study nor were the negative reactions to the LEAN management implementation, or LEAN management as such. The engaging leadership behaviors that were taught throughout the leadership intervention as described in Figure 1, were, however, meant to help turn the negative sentiment around so that, for example, the continuous improvement conversations were no longer seen and an obligation, but as a space for learning and growth. Which is also why senior leadership came up with the slogan “co-creating a great place to work.”

The reason for not describing these aspects in the present paper was that the study was not about LEAN management. This would have distracted from the subject of engaging leadership, need satisfaction and the intervention study itself. Also, it would have had considerable consequences for the length of the paper. However, we are of course willing to add extra detail if the reviewer deems it absolutely necessary.

Reviewer 2 Report

Main Comments and Suggestions

First, I am not sure the particular leadership program is specifically designed for the purpose. In other words, can the research results be generalized for other firms, industries, and countries? You should clarify the contributions of the paper which are not elaborated well in the current paper. You can talk about the following contributions: What insights can you provide based on your finding? Do they push forward our understanding? What should we do with your research? Do you have any suggestions to improve the current regulation or practice? Adding the above discussion and extend your literature review may help you make more contributions and position your contributions better.

The results can be driven by unobservable CEO characteristics you need to discuss. See Coles and Li, 2019. An Empirical Assessment of Empirical Corporate Finance. In addition, you should acknowledge that this is not a double blind experiment where the participants do not know the experiment. The survey should be provided as appendix and more information about the experiment should be discloses, so readers learn what (and what not) can be used in their situations.

Minor Comments and Suggestions

There are grammatical mistakes and awkward sentences throughout the paper, making it hard to read and understand. Try to avoid long sentences and vague words. Use short, precise, and concise sentences and be more straightforward. The last section of conclusion, which is too short, should summarize all your findings, their implications to researchers and practitioners, future direction for research, limitation of the current study, etc. You need to seriously proofread the paper and extend and update your references.

In conclusion, I would like to thank the authors for a very interesting, unique and potentially important paper. Hope these comments and suggestions can help further their study.

Author Response

RV (Reviewer): First, I am not sure the particular leadership program is specifically designed for the purpose. In other words, can the research results be generalized for other firms, industries, and countries?

You should clarify the contributions of the paper which are not elaborated well in the current paper. You can talk about the following contributions: What insights can you provide based on your finding? Do they push forward our understanding? What should we do with your research? Do you have any suggestions to improve the current regulation or practice? Adding the above discussion and extend your literature review may help you make more contributions and position your contributions better.

AW (Answer): We have added a new section on the contributions of the study at line 475 (Track Changes off): 4.2. Contribution to knowledge development

The present study is the first in the domain of SDT  to report the impact of a leadership development intervention on actual business performance, as measured with business key performance indicators (i.e., productivity and absenteeism). Previous studies published on leadership impact within SDT reported on, for example, performance evaluations by supervising managers [39] or self-reported in-role versus extra-role behaviors [38]. Another study measured the impact of autonomy-supportive leadership on employee perceptions [34], and Deci et al. [33] reported on the positive effects of a leadership intervention on supervising managers' orientations. Similar to the present study also in Deci et al.'s research, the impact on employees was less conclusive. The present study's contribution to leadership and SDT is relevant because it adds to the range of beneficial results that may be obtained through engaging leadership behaviors (connecting, empowering, strengthening) [25] aiming to satisfy basic psychological needs [13]. It also substantiates the claim that basic need satisfaction through autonomy-supportive leadership behaviors, such as described in engaging leadership, positively associates with enhanced performance [11,65,66].

Moreover, the present study also contributes to leadership development. The design of the leadership intervention facilitated the collective leadership experience through the participatory process of co-creation of senior leaders and the participating team leaders, and hence followed the call of Petri [67] for more collective, networked approaches to leadership development. Engaging the participating team leaders supported the inclusion of the selected business KPI, absenteeism, and motivation as pillars of the progamme. Furthermore, engaging leadership behaviors [25] that were taught in the training sessions followed the principles of coaching for leadership development, as described by Ting [68]. Examples are: creating a safe and challenging environment (connecting), choosing facilitative leadership over directive action planning (empowering), and promoting learning through collectively evaluating experiences and applying these to improve the business outcomes further (strengthening).

We believe that designing a leadership development programme in a real-time work environment with a real business challenge [cf. 69] answers to the need for more contextually embedded leadership initiatives [70] and measurable outcomes [71]. We suggest that future leadership development practice can benefit from the incorporation of leadership development programmes – like the one in the current study – in the day-to-day processes of departments and organizations [3].

RV: The results can be driven by unobservable CEO characteristics you need to discuss. See Coles and Li, 2019. An Empirical Assessment of Empirical Corporate Finance

AW: That was a very interesting read! Thank you very much for bringing this to our attention. We have added the following to the limitations section:

Added around line 544 (track changes off): “Another limitation on causality is the potential impact of unobserved managerial and organizational effects on observed performance, which may explain additional the variance in the outcome variables [74]. We could not control for such influences.

RV: In addition, you should acknowledge that this is not a double-blind experiment where the participants do not know the experiment.

AW: We have added a fourth limitation in the first paragraph of the limitation section to address this issue. 

Added around line 529 (track changes off) “Fourth, the quasi-experimental design plus the co-creation phase of the program, in which the senior leadership and team leaders participated, may have drip-fed information on the proposed leadership intervention to team members previous tothe initial survey. Hence, it can not be assumed that the intervention group’s team members were completely unaware of the experiment when filling out the survey. This limitation did, however, not extend team members of to the control group because only its head of department knew about the experiment in the intervention group.”

RV: The survey should be provided as appendix and more information about the experiment should be discloses, so readers learn what (and what not) can be used in their situations.

AW: We have added the survey items as appendix A at line 595 (With track changes off). The business performance parameters and absenteeism were not survey items.

Survey items

Engaging Leadership

Select the comment that best describes your present agreement or disagreement with each statement below. At work my supervisor…

Strengthening

1

encourages team members to develop their talents as much as possible

2

delegates tasks and responsibilities to team members

3

encourages team members to use their own strengths

Connecting

4

encourages collaboration among team members

5

actively encourages team members to aim for the same goals

6

promotes team spirit

Empowering

7

gives team members enough freedom and responsibility to complete their tasks

8

encourages team members to give their own opinion

9

recognises ownership of team member’s contributions

Autonomy satisfaction

At work…

10

I am free to do things my own way

11

My choices express my ‘‘true self’’

12

I am really doing what interests me.

Intrinsic motivation

Why do you or would you put efforts in your job?

13

Because I have fun doing my job

14

Because what I do in my work is exciting

15

Because the work I do is interesting

RV: There are grammatical mistakes and awkward sentences throughout the paper, making it hard to read and understand. Try to avoid long sentences and vague words. Use short, precise, and concise sentences and be more straightforward.

AW: The entire manuscript was checked once more on English language style and grammatical errors by the first author and a native speaker. Many sentences were shortened, passsages were rewritten and made more accessible. Also, punctuation was checked. The paragraph about implications for practitioners was thoroughly rewritten.

RV: The last section of conclusion, which is too short, should summarize all your findings, their implications to researchers and practitioners, future direction for research, limitation of the current study, etc.

You need to seriously proofread the paper and extend and update your references.

AW: We have rewritten the conclusion section (Section 5, page 14), line 556 (track changes off). Additionally, we have checked, extended, and updated the references.

5. Conclusion

The present study showed that a leadership development programme focusing on engaging leadership and psychological well-being led to significant positive business results and lower absenteeism. Additionally, the intervention had a moderate to large effect for the motivation of team leaders and a substantial impact on the experienced autonomy of team leaders compared to the control group. The business outcomes exceeded the leadership team’s ambitions and continued to improve post-programme. The predicted positive spill-over effects on team members were absent despite the positive psychological effects on autonomy and motivation among team leaders and their subsequent well-being.

From a practical point of view, co-creating a leadership programme by actively involving its participants helped create direction, alignment, and commitment. It also supported the realization of business outcomes in a spirit of collaboration and continuous improvement. Embedding leadership development in everyday business processes with concrete business objectives and extending co-creation throughout the programme supports team leaders in developing engaging leadership in their day-to-day practice and contributes to tangible results. The next step is to find ways to extend the programme and engage team members in co-creation and implementation. Considering the current trend of self-management and agility in business organizations, we argue that leadership development should not be limited to management or a select group of high-potential individuals.

Despite the positive results of the programme, generalization across organizations is limited, because it comprised a single intervention in one organization and a relatively small group of team leaders. Additionally, co-creation may lead to different foci in different organizations and influence the outcomes. However, we believe that the described approach can be strengthened through replication in other organizations. The present study contributes to the practical knowledge of the relatively new concept of engaging leadership and its successful application in leadership development programmes.

RV: RV: In conclusion, I would like to thank the authors for a very interesting, unique and potentially important paper. Hope these comments and suggestions can help further their study.

AW: Thank you very much for your comments and the time invested. We very much appreciate it.

Reviewer 3 Report

    This manuscript tests the business impact of the leadership development plan. Using the concepts of participatory leadership and self-determination theory, the eight-month plan targets the middle-level team leader of the customer fulfillment center of a health systems multinational organization. The plan was created by senior leadership and the team leaders that participated in the programme. The main purpose of this study is to improve objective business performance and reduce absenteeism through leadership development programs. The second purpose is to test whether the content and design of the leadership plan will benefit the team leaders participating in the plan. This study helps to understand the relatively new concept of leadership and its application in leadership development programs.

    This article mentions "To the best of our knowledge, no studies using SDT...", but the description of this contribution is not specific enough. It is recommended to confirm the importance of this contribution in the research results. In addition, this experimental study only uses the mean and standard deviation to compare the experiment with the control group. It seems that the degree of research is not deep enough. It is recommended to conduct a robustness test to improve the credibility of the study.

Author Response

RV (Reviewer): This manuscript tests the business impact of the leadership development plan. Using the concepts of participatory leadership and self-determination theory, the eight-month plan targets the middle-level team leader of the customer fulfillment center of a health systems multinational organization. The plan was created by senior leadership and the team leaders that participated in the programme. The main purpose of this study is to improve objective business performance and reduce absenteeism through leadership development programs. The second purpose is to test whether the content and design of the leadership plan will benefit the team leaders participating in the plan. This study helps to understand the relatively new concept of leadership and its application in leadership development programs.

RV: This article mentions "To the best of our knowledge, no studies using SDT...", but the description of this contribution is not specific enough. It is recommended to confirm the importance of this contribution in the research results.

AW (Answer): We replaced the “To the best of our knowledge, no studies using SDT include actual business performance as an outcome” (line 118, track changes off) with: “We have found no studies within the domain of SDT reporting on actual business performance as an outcome measure.”

We assume this will make it more specific. To be sure, we have checked anew whether we could find any studies within SDT reporting on actual business performance but, indeed, we have found none.

In response of the comments of another reviewer we added a new paragraph in the discussion section under the header “4.2 Contribution to knowledge development” (starting at line 474, track changes off)

The part relevant to the feedback on contribution is pasted below. The full passage can be read in the document.

Tex added:

4.2. Contribution to knowledge development

In the domain of SDT, the present study was the first to report the impact of a leadership development intervention on actual business performance, measured through a business key performance indicator on productivity and absenteeism. Previous studies published on leadership impact within SDT reported on, for example, performance evaluations by supervising managers [39] or in-role versus extra-role behaviors [38]. Another study measured the impact of autonomy-supportive leadership on employee perceptions [34], and Deci et al. [33] reported on the positive effects of a leadership intervention on supervising managers' orientations. Similar to the present study also in Deci et al.'s research, the impact on employees was less conclusive. The present study's contribution to leadership and SDT is relevant because it adds to the range of beneficial results that may be obtained through engaging leadership behaviors (connecting, empowering, strengthening) [25] aiming to satisfy basic psychological needs [13]. It also substantiates the claim that basic need satisfaction through autonomy-supportive leadership behaviors, such as described in engaging leadership, positively associates with enhanced performance [11,65,66]

RV: In addition, this experimental study only uses the mean and standard deviation to compare the experiment with the control group. It seems that the degree of research is not deep enough. It is recommended to conduct a robustness test to improve the credibility of the study.

AW: To answer to your concern we have reached out to a statistician at the department of methodology and statistics at Utrecht university, who was also consulted on previous drafts of the present study. Initially, we considered a non-parametric test, but there is no straightforward one in case of conditioning on covariates and correcting for multiple testing (because of the 3 dependent variables) and taking into account the dependency of the dependent variables. Therefore, we performed a MANCOVA, which also increases power. We checked the assumptions, as described in the paper, and then ran the test. Due to the small sample size and thus low power, we solely inspected effect sizes and not parameter estimates (with their standard errors). We opted for partial epsilon^2, because it is a more reliable effect size in small samples (Okada, 2013; Albers & Lakens, 2018)

We aimed to obtain more robust, bias-corrected confidence intervals for the parameter estimates and the effect sizes by using bootstrap (for which we used R). Nevertheless, because of the small sample, the intervals for the estimates still contain zero and those for the effect sizes are wide (which is to be expected because bootstrapping does not increase power). We saw no further possibilities to test for robustness.

This is the first study into the impact of engaging leadership and need satisfaction in an intervention study. In a following study, it would be great to have a larger sample size of team leaders such that relevant differences can become significant and robust, and bias-corrected confidence intervals more meaningful.

Because of this comment, we added a sentence to the limitations section around line 528 (Track changes off) after “Third, the study incorporated small groups of team leaders, and the results could not be nested because of the survey protocol of the organization.”  

We added: “Because of the small sample size, the power was too low to obtain significant parameter estimates and relied on effect sizes only. We selected the partial epsilon squared because it is said to be the least biased in small samples [58]. A future intervention study would benefit from larger groups of team leaders, which makes significance and robustness of relevant differences more feasible.”

For the significant work the statistician has invested this week to rebuild the complete analyses in R in we have added her as a co-author.

Round 2

Reviewer 2 Report

Well done. Congrats!

Reviewer 3 Report

This manuscript has been revised in accordance with the recommendations, and its significance for research and practice has been determined. The manuscript also contains important information sufficient to justify its publication.